# Control of AMPA receptor activity by the extracellular loops of auxiliary proteins

**Irene Riva[1,2†], Clarissa Eibl[1,2†], Rudolf Volkmer[3], Anna L Carbone[1,2*], Andrew JR Plested[1,2*]**

[1]Institute of Biology, Cellular Biophysics, Humboldt Universität zu Berlin, Berlin, Germany; [2]Molecular Physiology and Cell Biology, Leibniz-Forschungsinstitut für Molekulare Pharmakologie, Berlin, Germany; [3]Chemical Biology, Leibniz-Forschungsinstitut für Molekulare Pharmakologie, Berlin, Germany

**Abstract** At synapses throughout the mammalian brain, AMPA receptors form complexes with auxiliary proteins, including TARPs. However, how TARPs modulate AMPA receptor gating remains poorly understood. We built structural models of TARP-AMPA receptor complexes for TARPs γ2 and γ8, combining recent structural studies and de novo structure predictions. These models, combined with peptide binding assays, provide evidence for multiple interactions between GluA2 and variable extracellular loops of TARPs. Substitutions and deletions of these loops had surprisingly rich effects on the kinetics of glutamate-activated currents, without any effect on assembly. Critically, by altering the two interacting loops of γ2 and γ8, we could entirely remove all allosteric modulation of GluA2, without affecting formation of AMPA receptor-TARP complexes. Likewise, substitutions in the linker domains of GluA2 completely removed any effect of γ2 on receptor kinetics, indicating a dominant role for this previously overlooked site proximal to the AMPA receptor channel gate.

DOI: https://doi.org/10.7554/eLife.28680.001

**\*For correspondence:**
carbone@fmp-berlin.de (ALC);
plested@fmp-berlin.de (AJRP)

[†]These authors contributed equally to this work

**Competing interests:** The authors declare that no competing interests exist.

## Introduction

Since the identification of the protein Stargazin, also known as γ2, as the prototype transmembrane AMPA receptor regulatory protein (TARP) (*Chen et al., 2000*), several families of auxiliary proteins for the AMPA receptor have been described that include TARPs (*Schwenk et al., 2012*; *Tomita et al., 2003*), cornichons (*Schwenk et al., 2009*), GSG1L (*Shanks et al., 2012*) and CKAMPs (*von Engelhardt et al., 2010*; *Klaassen et al., 2016*). These proteins play an essential role in tethering AMPA-type glutamate receptors at the synapse, and also exert complex control over surface expression of functional receptors (*Dakoji et al., 2003*; *Yamazaki et al., 2010*). Auxiliary proteins regulate the function of AMPA receptors, with both positive and negative modulation of gating (*von Engelhardt et al., 2010*; *Priel et al., 2005*; *McGee et al., 2015*; *Rouach et al., 2005*), as well as control over permeation and block (*Soto et al., 2007*). The range of auxiliary subunit influence over synaptic transmission is compounded by striking regional and cell-type specific expression (*Tomita et al., 2003*; *Kato et al., 2007*), and a patchwork of interaction patterns (*Bats et al., 2012*; *Kato et al., 2010*).

TARPs and other auxiliary proteins modify the gating and pharmacology of synaptic AMPA receptors (*Boudkkazi et al., 2014*; *Milstein et al., 2007*). The physiological importance of modulation is likely to be the specialization of particular codes of short-term plasticity, in the hippocampus and cerebellum at least (*von Engelhardt et al., 2010*; *Klaassen et al., 2016*; *Khodosevich et al., 2014*; *Devi et al., 2016*). Recently, antagonists of AMPA receptors that target GluA2–γ8 complexes were described (*Maher et al., 2016*; *Kato et al., 2016*), further enhancing interest in the molecular basis of complexes of GluA subunits and their auxiliary proteins as potential drug targets.

Previous studies showed that some of the effects of auxiliary proteins on receptor gating were due to the extracellular domains (*Tomita et al., 2005*; *Tomita et al., 2007*; *Cho et al., 2007*). However, several of these studies made use of chimeras with γ5, which was presumed to be a null subunit, but which was subsequently shown to modulate gating and conductance of GluA receptors (*Soto et al., 2009*). The results obtained from these studies are therefore difficult to interpret because modulatory effects observed could have been mediated by either of the TARPs forming the chimera. Although some mutations in extracellular portions of TARPs were reported that affect TARP activity, there is no clear indication that these TARPs formed complexes with GluA subunits as well (*Cais et al., 2014*). On the other hand, some studies of assembly made use of functional tests to assess the strength of interaction (*Shi et al., 2009*). Given the variable stoichiometry of assembly between different TARP isoforms (*Kim et al., 2010*; *Hastie et al., 2013*), interpreting these data, which combine the strength of association, expression and modulation into a single metric, is difficult. Very recently, a chimeric approach confirmed impressions from structural studies that transmembrane interactions are important for proper assembly, with the TM3 and TM4 segments of γ2 and the M1-M3 helices of the AMPA receptor determining complex assembly. However, the C-termini of both the AMPA receptor and TARPs also appear to be involved (*Ben-Yaacov et al., 2017*). Despite these insights, there is very little information about the extent to which different domains contribute to gating of complexes (*Twomey et al., 2017a*), and no information about the structural basis of slow modulation, superactivation (*Carbone and Plested, 2016*).

Two of the predominant TARPs in the brain are the auxiliary proteins γ2 and γ8. In this work, we isolate the extracellular segments of γ2 and γ8 that are responsible for modulation of gating, and show that these segments act on the receptor via the linkers connecting the ligand binding domain (LBD) and the transmembrane domain (TMD). In so doing, we were able to produce 'null' TARPs, which assemble normally but show no modulation of gating. Hereby, we establish mechanisms for the subunit specific modulation of AMPA receptors by auxiliary proteins.

## Results

### A model of auxiliary protein interactions

Previous studies of TARP modulation of AMPA receptors have identified extracellular regions as potential interaction motifs. Crystal structures of Claudins, proteins with close homology to TARPs, enabled a more refined view, defining a folded extracellular 'cap' (*Suzuki et al., 2014*; *Saitoh et al., 2015*; *Shinoda et al., 2016*) that substantially limits the sections of the extracellular portion of TARPs that are able to interact with the AMPA receptor, and therefore the likely range of these interactions. More recently, cryo-EM/single particle analysis of GluA2-TARP complexes allowed unambiguous positioning of TARPs at the periphery of the GluA2 pore, and partially resolved the extracellular domains of TARPs (*Twomey et al., 2016*; *Zhao et al., 2016*). The major sequence and structural differences between Claudin and TARP proteins, and between TARPs with different modulatory effects, are found in the variable extracellular loops between β1 and β2 (Loop 1), and between TM3 and β5 (Loop 2, *Figure 1A*). We sought to identify interactions between TARPs and the extracellular regions of the GluA2 receptor on this basis.

To understand the scope of TARP interactions with the AMPA receptor, we began by modeling the loops of γ2 and γ8 into a hybrid structure composed of Claudins and GluA2. Based on the crystal structure of the related claudin15 (PDB code: 4P79) we first generated structural models of TARP γ2 and γ8 including the presumably flexible loop 1, which is not resolved in the cryo-EM AMPA-γ2 complex structures (PDB code: 5KBU and 5KK2). To ensure that the models obeyed good stereochemistry, we analyzed them with MolProbity (*Chen et al., 2010*). The analysis of the in silico models revealed bad bonds and angles of 0.3% and 1.4% for γ2 and 0.2% and 1.6% for γ8. These TARP models were superposed onto the γ2 chains in the cryo-EM GluA2-γ2 complex (PDB code: 5KK2). L1 had to be repositioned in order to avoid a steric clash with the LBD. We show two extreme conformations of L1 in *Figure 1B*, purely to illustrate the reach of L1, which seems likely a function of its length (compare L1 of γ2 and γ8). The principal advantage of building these models was to enable us to concentrate on a range of physically-plausible interactions at the linkers and the lower regions

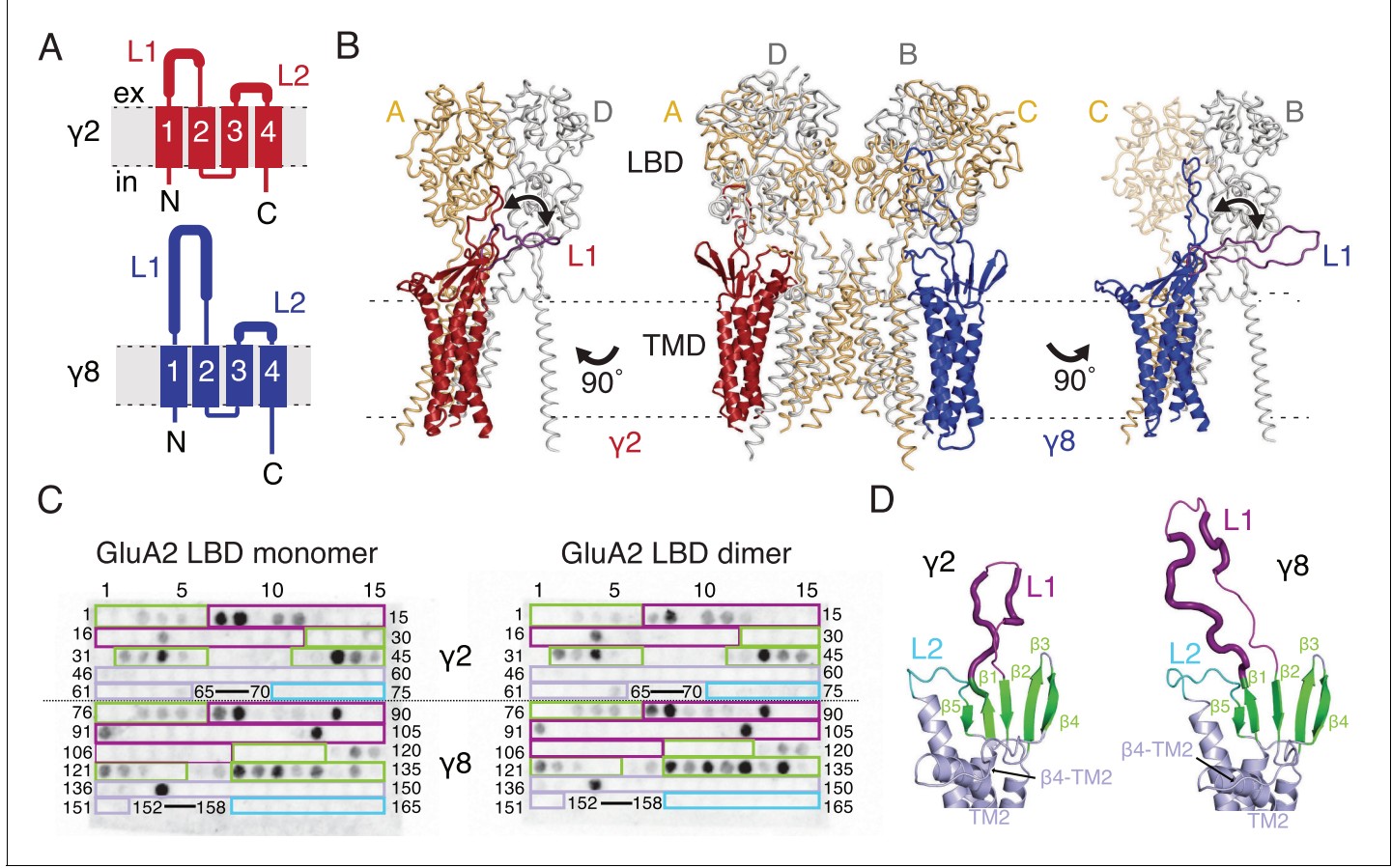

**Figure 1.** Modeling and biochemical analysis of AMPA-TARP complexes. (**A**) Topology of TARP γ2 (red) and γ8 (blue). Membrane helices numbered from 1 to 4. The first extracellular segment includes a flexible loop (L1, thick section, longer in γ8). Transmembrane helices 2 and 3 are connected by loop 2 (L2) (**B**) The middle panel shows TARPs γ2 (*red*) and γ8 (*blue*) positioned between equivalent receptor subunits (A and D and B and C) based on the cryo-EM complex structure (5kk2). The predicted L1 of γ8 is longer than in γ2 enabling it to reach more extensive regions of the receptor. To account for its flexibility we modeled L1 in two extreme positions (indicated by the double-headed arrows), either between the LBD dimer (colored like the respective TARP) or underneath the lower lobe of the LBD (*purple; left panel* for γ2, *right panel* for γ8). Potential L1 interactions with the LBD depend on its location in the complex (for example, between subunits A and B or A and D; see ***Figure 1—figure supplement 1A***). (**C**) Membranes spotted with overlapping hexameric peptides of the extracellular segments of γ2 (1–75) and γ8 (76–165) were incubated with either monomeric (*left*) or dimeric His-tagged GluA2 LBD (*right*). Interacting peptides give a dark spot on the membrane (darker spots indicate stronger binding) when developed with an HRP-conjugated antibody against the His-tag (see Materials and methods). Spots from 66 to 69 and 153–157 did not contain any peptides. The colored boxes indicate peptides location, with β-sheets (green), loop1 (purple), β4-TM2 loop containing the sequence of the negative patch (light blue) and L2 (cyan). See also panel D and ***Figure 1—figure supplement 1C***. Quantitation of the spot arrays is found in ***Figure 1—source data 1***. (**D**) Close up view on the modeled extracellular region of γ2 (*left*) and γ8 (*right*). Secondary structure elements are shown in cartoon representation in the same color code as in panel C. Positive peptide hits in L1 are indicated by thicker loop-representation.

DOI: https://doi.org/10.7554/eLife.28680.002

The following source data and figure supplements are available for figure 1:

**Source data 1.** Spot array quantitation.
DOI: https://doi.org/10.7554/eLife.28680.007
**Figure supplement 1.** Loop interactions between TARPs and GluA2.
DOI: https://doi.org/10.7554/eLife.28680.003
**Figure supplement 2.** The negatively charged patch on β4-TM2 loop of γ2 negatively modulates AMPA receptor gating.
DOI: https://doi.org/10.7554/eLife.28680.004
**Figure supplement 2—source data 1.** Rectification indices for negative patch chimera.
DOI: https://doi.org/10.7554/eLife.28680.008
**Figure supplement 3.** Sequence alignment and conservation of TARP loop 2.
DOI: https://doi.org/10.7554/eLife.28680.005
**Figure supplement 4.** Sequence alignment of γ2 and γ8 constructs.

*Figure 1 continued*

DOI: https://doi.org/10.7554/eLife.28680.006

of the LBD layer, and effectively rule out ATD contacts, because the unstructured loops are too short.

Comparing these hybrid models to CryoEM electron density maps suggested that a range of interaction sites with the LBD-TMD linkers and D2 domains of the LBD are possible (*Figure 1B* and *Figure 1—figure supplement 1A*). Whereas TARP loop 2 (L2) could engage in the receptor's pore four-fold symmetry, loop 1 (L1) reaches up to the two-fold symmetry of the LBD layer. In other words, while L2 can interact four times in the same way with the receptor (*Figure 1—figure supplement 1B*), L1 has at least two distinct modes of interaction depending on to which receptor subunits the TARP is adjacent (subunit A-D and B-C, *Figure 1B*, or A-B and C-D, *Figure 1—figure supplement 1A*). The variable loop 1 is not resolved in structures to date, consistent with it being a flexible modulatory element. Superactivation of GluA2 receptors resembles strongly the slow modulation of AMPA receptors by particular allosteric modulators that bind at the dimer interface (*Kato et al., 2010*; *Carbone and Plested, 2016*). We reasoned that extracellular loop interactions that stabilized the superactive state could preferentially target the GluA2 LBD dimer. To test this hypothesis of direct interactions between loop 1 and the GluA2 LBDs, we composed an overlapping library of hexameric peptides encompassing the extracellular sections of TARPs, targeting primarily the long loop 1 of γ2 and γ8, and other potential interacting sites (*Figure 1C and D* and *Figure 1—figure supplement 1C*). Because the active dimer of LBDs ought to be intact for superactivation, we compared the interactions of our peptide library, which was spotted onto cellulose membranes and then either incubated with the monomeric GluA2 LBD (flip form) or LBDs harboring the L483Y substitution, which greatly increases dimer formation in solution (compare left and right panel in *Figure 1C*).

Repeated peptide mapping array assays indicated no clear preference for either monomeric or dimeric GluA2 LBD. However, in accordance with our hypothesis the majority of the L1 of both γ2 and γ8 contain hits (dark spots within the purple boxes, *Figure 1C*) in the peptide mapping array, indicating direct interaction with the receptor LBD (*Figure 1C and D*, *Figure 1—figure supplement 1C*), albeit in conditions lacking the usual steric constraints of the complex. In the recent cryo-EM structures of the GluA2-TARP complex a possible interaction between a conserved negatively charged region (negative patch, NP) located on the TARP β4-TM2 loop and the KGK motif in the lower lobe of the GluA2 LBD was predicted (*Zhao et al., 2016*; *Twomey et al., 2016*). Thus we also tested for this potential interaction in the peptide mapping array but found no hits. A functional test of neutralizing three of the six acidic residues in this patch (D88G, E90S, D92G) made γ2 into a much stronger modulator of AMPAR gating, with the steady-state current and superactivation both doubled (*Figure 1—figure supplement 2*). This result suggested that if interactions of the TARP negative patch with the receptor alter function, they actually inhibit the action of γ2. However, other sites have a dominant effect in the positive modulation of gating.

We also tested L2 of γ2 and γ8 in the peptide mapping array for possible interactions with the LBDs because of its conserved charged features (4 and 7 charges), which are less prominent in γ5 and γ7 (3 and 1 charges respectively) (*Figure 1—figure supplement 3*). Considering L2 being positioned distant underneath the LBD (around 15 Å, measured between Cα of GluA2 Pro717 and γ2 Lys170 in the complex from PDB code: 5KBU [*Twomey et al., 2016*]) in the cryo-EM structures, it was not surprising that we found no interaction between L2 peptides and the GluA2 LBD. According to our GluA2-TARP models, in both γ2 and γ8, L2 is adjacent to the receptor S1-M1 and S2-M4 linkers (*Figure 1—figure supplement 1A and B*), which are outside the realms of our LBD construct.

## Modulation of fast AMPA receptor gating by TARP L1 and L2 segments

To investigate the role of the extracellular domain of TARPs in controlling AMPA receptor activation, we made a series of chimeras and deletion mutants between γ2 and γ8. We first targeted the long loop in the first extracellular segment L1 (*Figure 1*) that has markedly different lengths and sequence content across the TARP family and its homologs. We also investigated the role of the shorter unstructured region in the second extracellular segment L2 (*Figure 1*), which is poised to interact with the LBD-TMD linkers of the AMPA receptor.

We first swapped L1 between γ2 and γ8 (*Figure 2A* and *Figure 1—figure supplement 4*), and assessed effects on desensitization. Although γ2 and γ8 apparently affect AMPA receptor desensitization similarly, γ8 slows down entry to desensitization more than γ2 ($60 \pm 5$ $s^{-1}$ and $40 \pm 5$ $s^{-1}$, $n = 24$ and 9, for γ2 and γ8, respectively; *Table 1*). These chimeras exhibited asymmetric effects on desensitization. When activated by 10 mM glutamate, the chimera of γ2 with L1 from γ8 had steady-state current of $50 \pm 5\%$ ($n = 30$; *Figure 2A and D* and *Table 1*), twice as large as γ2 alone ($25 \pm 2\%$, $n = 24$ patches), and the rate of entry to desensitization was approximately halved ($35 \pm 5$ $s^{-1}$,

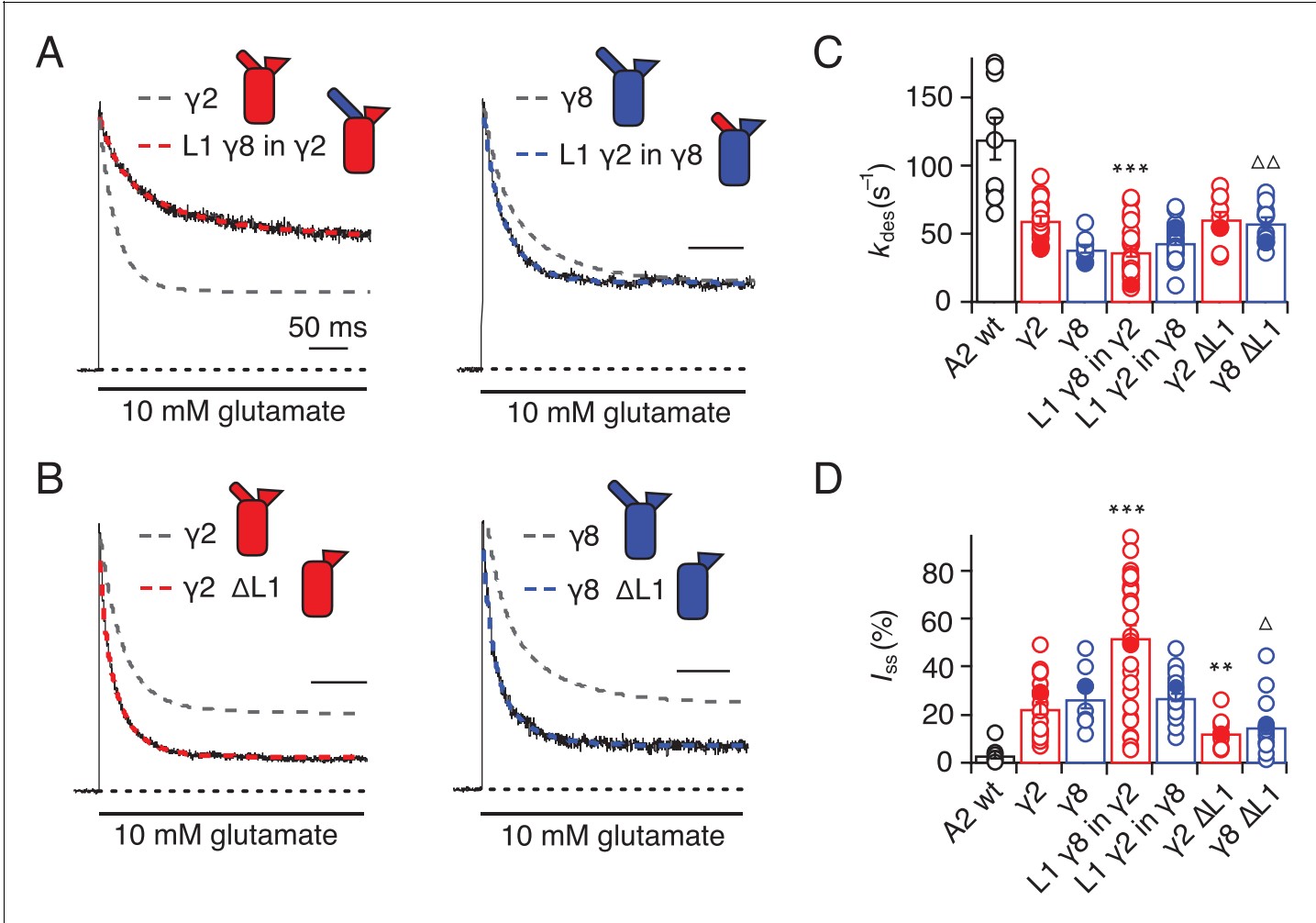

**Figure 2.** Desensitization properties of γ2 and γ8 L1 mutants. (**A**) Representative traces from L1 γ8 in γ2 (*red*) and L1 γ2 in γ8 (*blue*) coexpressed with GluA2 in response to a 500 ms pulse of 10 mM Glutamate ($k_{des}$ = 13 and 55 $s^{-1}$; $I_{ss}$ = 50% and 30%, respectively). Example traces recorded from the parent TARPs coexpressed with GluA2 are shown in grey for comparison ($k_{des}$ = 41 and 30 $s^{-1}$; $I_{ss}$ = 30% and 30%, for γ2 and γ8, respectively). (**B**) Representative traces from γ2 ΔL1 (*red*) and γ8 ΔL1 (*blue*) coexpressed with GluA2 in response to a 500 ms pulse of 10 mM Glutamate ($k_{des}$ = 55 and 45 $s^{-1}$; $I_{ss}$ = 10% and 15%, respectively). The wild type constructs coexpressed with GluA2 are shown as dashed lines for comparison. (**C**) Bar graph summarizing the effects of the L1 mutation on the desensitization kinetics. (**D**) Bar graph summarizing the effects of the loop1 mutations on the steady state current of the complexes. Currents were recorded at +50 mV in the presence of 50 μM spermine in the pipette solution. For panels C and D, filled symbols correspond to the traces shown in (**A**) and (**B**). ***$p<0.001$, **$p<0.01$, against γ2; Δ $p<0.05$, ΔΔ $p<0.01$, against γ8. Source data for panels C and D is found in *Table 1–source data 1*. Error bars represent s.e.m.

DOI: https://doi.org/10.7554/eLife.28680.011

The following source data and figure supplements are available for figure 2:

**Figure supplement 1.** Relief of polyamine block is not affected by loop mutations in γ2 and γ8.

DOI: https://doi.org/10.7554/eLife.28680.012

**Figure supplement 1—source data 1.** Rectification indices for electrophysiological recordings of TARP chimeras.

DOI: https://doi.org/10.7554/eLife.28680.013

**Table 1.** Kinetic properties of wild type and chimeric TARPs and GluA2 linker mutants.

$k_{des}$ is rate of desensitization, $I_{ss}$ the steady state current expressed as percentage of the peak current and "Superact." the extent of superactivation expressed as the slow increase in steady state current during prolonged exposure to glutamate (see Materials and methods for details). The number of patches recorded for each condition is shown in brackets. Values are shown as mean ± s.e.m. $p$ values (from Student's $t$ test) are calculated as follows: § against the parent TARP; Δ against GluA2 WT; # against GluA2 WT + TARP. Currents recorded in the presence of TARPs were held at +50 mV in the presence of 50 µM spermine in the pipette solution. Recordings in the absence of TARPs were done at –60 mV without intracellular polyamines. Source data for **Table 1** is found in **Table 1– source data 1**.

| Construct | $k_{des}$ (s-1) | P | $I_{ss}$ (%) | P | Superact. (%) | P |
|---|---|---|---|---|---|---|
| A2 wt | 120 ± 15 (9) | | 5 ± 1 | | — | — |
| γ2 | 60 ± 5 (24) | | 25 ± 2 | | 7 ± 2 (10) | |
| γ8 | 40 ± 5 (9) | | 25 ± 5 | | 30 ± 6 (4) | |
| γ2 β4 TM2 § | 40 ± 5 (7) | 0.004 | 50 ± 5 | $1 \times 10{-5}$ | 17 ± 4 (5) | 0.009 |
| L1 γ8 in γ2 § | 35 ± 5 (30) | $5 \times 10{-6}$ | 50 ± 5 | $7 \times 10{-6}$ | 27 ± 6 (10) | 0.003 |
| L1 γ2 in γ8 § | 45 ± 1 (28) | 0.34 | 25 ± 3 | 0.86 | 16 ± 1 (16) | 0.001 |
| γ2 ΔL1 § | 60 ± 5 (11) | 0.90 | 15 ± 2 | 0.008 | 6 ± 2 (6) | 0.52 |
| γ8 ΔL1 § | 60 ± 5 (15) | 0.002 | 15 ± 3 | 0.03 | 16 ± 3 (6) | 0.02 |
| γ2 L2_GS § | 65 ± 5 (15) | 0.49 | 5 ± 1 | $1 \times 10{-6}$ | 1.3 ± 0.6 (8) | 0.003 |
| γ8 L2_GS § | 25 ± 5 (6) | 0.002 | 40 ± 4 | 0.07 | 12 ± 2 (4) | 0.01 |
| L1 γ8 in γ2 L2_GS § | 10 ± 0.5 (7) | $6 \times 10{-10}$ | 45 ± 3 | $6 \times 10{-5}$ | 4 ± 2 (6) | 0.19 |
| L1 γ2 in γ8 L2_GS § | 85 ± 5 (6) | $1 \times 10{-5}$ | 5 ± 1 | 0.001 | 1 ± 0.7 (6) | $9 \times 10{-5}$ |
| γ2 ΔL1 L2_GS § | 80 ± 20 (5) | 0.03 | 2 ± 1 | $4 \times 10{-4}$ | 0 (4) | 0.011 |
| γ8 ΔL1 L2_GS § | 60 ± 10 (5) | 0.02 | 10 ± 5 | 0.02 | 3 ± 1 (4) | 0.02 |
| A2 K509A Δ | 100 ± 5 (5) | 0.34 | 3 ± 0.5 | 0.71 | — | — |
| A2 508GAG510 Δ | 145 ± 35 (3) | 0.42 | 1 ± 0.5 | 0.27 | — | — |
| A2 781GSG783 Δ | 110 ± 15 (3) | 0.76 | 2 ± 1 | 0.46 | — | — |
| A2 GAG/GSG Δ | 150 ± 20 (5) | 0.20 | 2 ± 1 | 0.44 | — | — |
| A2 K509A + γ2 # | 30 ± 10 (5) | $3 \times 10{-4}$ | 45 ± 3 | $2 \times 10{-4}$ | 5 ± 5 (4) | 0.59 |
| A2 508GAG510 + γ2 # | 70 ± 5 (4) | 0.39 | 10 ± 5 | 0.07 | 0 (3) | 0.03 |
| A2 781GSG783 + γ2 # | 60 ± 5 (9) | 0.60 | 10 ± 1 | 0.001 | 2 ± 0.5 (8) | 0.005 |
| A2 GAG/GSG + γ2 # | 80 ± 5 (8) | 0.01 | 5 ± 1 | $9 \times 10{-5}$ | 0 (4) | 0.01 |
| A2 GAG/GSG + L1 γ8 in γ2 # | 12 ± 0.5 (5) | $4 \times 10{-8}$ | 30 ± 5 | 0.21 | 2 ± 2 (4) | 0.065 |
| A2 GAG/GSG + γ8 # | 45 ± 2 (5) | 0.30 | 12 ± 3 | 0.03 | 25 ± 5 (5) | 0.37 |
| A2 GAG/GSG + L1 γ2 in γ8 # | 72 ± 5 (5) | $8 \times 10{-5}$ | 4 ± 1 | 0.001 | 1 ± 1 (4) | 0.001 |
| A2 GAG/GSG + γ2 L2_GS # | 90 ± 10 (9) | $3 \times 10{-4}$ | 2 ± 1 | $5 \times 10{-6}$ | 0 (4) | 0.01 |

DOI: https://doi.org/10.7554/eLife.28680.009

The following source data available for Table 1:**Source data 1.** Kinetics and steady state currents from electrophysiological recordings.

DOI: https://doi.org/10.7554/eLife.28680.010

$n = 30$; **Figure 2C** and **Table 1**). In contrast, the γ8 chimera with L1 from γ2 maintained the original desensitization behavior of the parent TARP ($45 \pm 1$ s$^{-1}$, $n = 28$; **Figure 2A and C** and **Table 1**). Deletion of L1 from γ2 and γ8 approximately halved the steady state current ($15 \pm 2$ and $15 \pm 3\%$, $n = 11$ and 15, for γ2 ΔL1 and γ8 ΔL1, respectively; **Figure 2B and D** and **Table 1**), with a barely detectable speeding up of entry to desensitization ($60 \pm 5$ s$^{-1}$, $n = 11$ and 15, for γ2 ΔL1 and γ8 ΔL1, respectively; **Figure 2B and C** and **Table 1**). These results suggested that L1 can influence desensitization of complexes, as shown recently for GSG1L (**Twomey et al., 2017a**) but the absence of a simple exchange in desensitization behavior suggested that this loop functions in concert with other modulatory elements.

Seeking a further explanation for the modulation of desensitization by TARPs, we investigated the effects of altering the 8-residue stretch in the second extracellular segment of TARPs (L2), which connects TM3 to β5 in the extracellular domain. Replacement of the L2 segment with a flexible Gly-Ser linker, predicted to be of sufficient length not to disrupt the overall structure of the extracellular domain, had a striking effect on γ2. The rate of entry to desensitization was still slower than in receptors formed of GluA2 wild type (WT) alone ($65 \pm 5\ s^{-1}$ and $120 \pm 15\ s^{-1}$, $n = 15$ and 9 patches for A2 + γ2 L2_GS and A2 WT, respectively; *Figure 3A and C* and *Table 1*), but the steady state current was reduced to the level of receptors without any TARP present ($5 \pm 1\%$ and $5 \pm 1\%$, $n = 15$ and 9 for A2 + γ2 L2_GS and A2 WT, respectively; *Figure 3A and D* and *Table 1*). In contrast, there was no detectable effect on γ8 of mutating this loop, except for a further slowing down of the desensitization rate ($k_{des} = 25 \pm 5\ s^{-1}$, $I_{ss} = 40 \pm 4\%$, $n = 6$, for γ8 L2_GS; *Figure 3A,C and D* and *Table 1*).

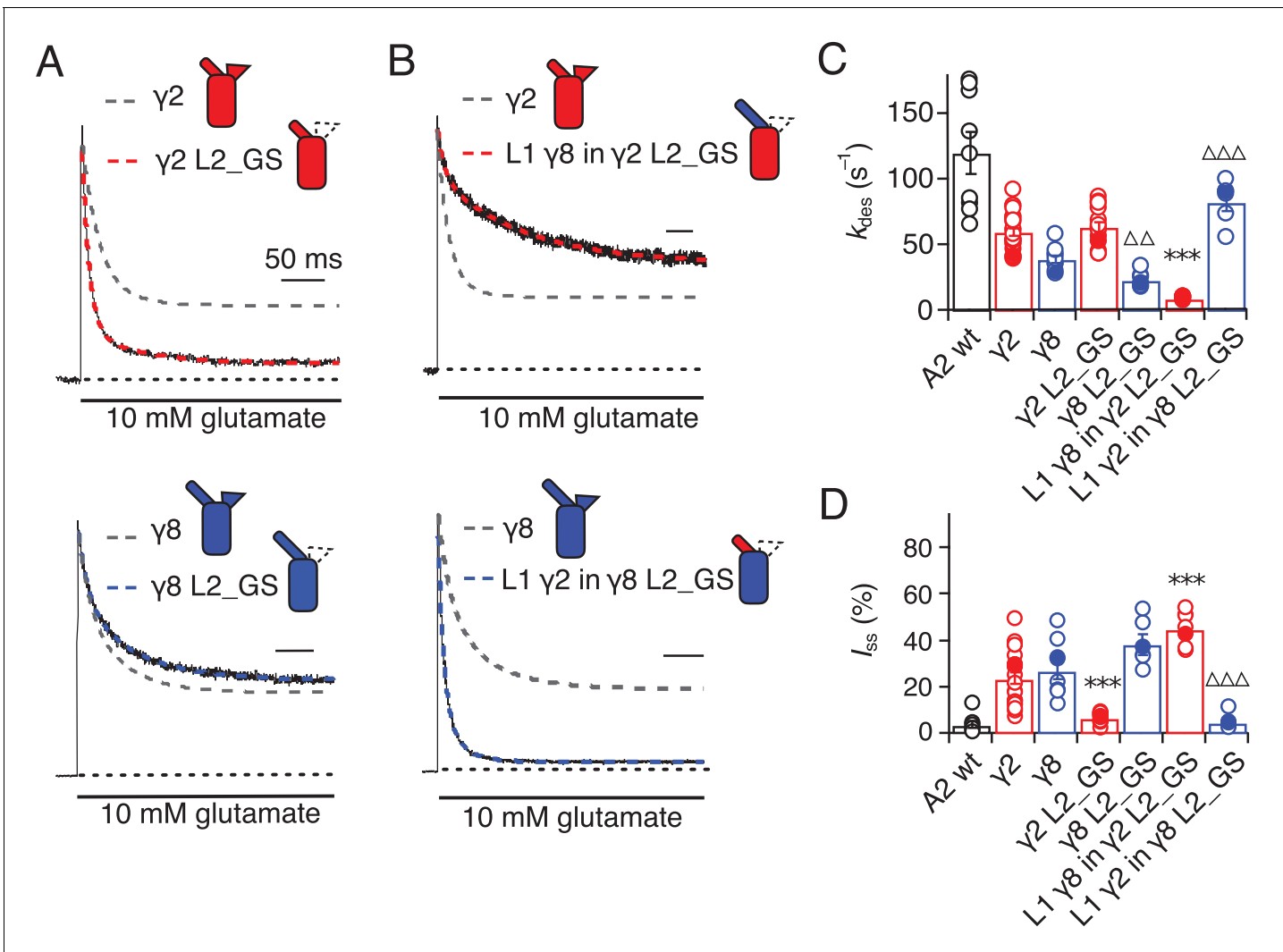

**Figure 3.** Desensitization properties of γ2 and γ8 L2 mutants. (**A**) Neutralization of L2 in γ2 (γ2 L2_GS, *red*) decreased $I_{ss}$, with little effect on γ8 (γ8 L2_GS, *blue*) ($k_{des} = 50$ and $20\ s^{-1}$; $I_{ss} = 5\%$ and 35%, respectively). Representative traces recorded from the parent TARPs are shown as dashed grey lines for comparison. (**B**) Representative traces from L1 γ8 in γ2 L2_GS (*red*) and L1 γ2 in γ8 L2_GS (*blue*) coexpressed with GluA2 in response to a 500 ms pulse of 10 mM Glutamate ($k_{des} = 10$ and $90\ s^{-1}$; $I_{ss} = 40\%$ and 5%, respectively). Traces from wild type γ2 and γ8 coexpressed with GluA2 are shown in grey for comparison. (**C**) Bar graph summarizing the effects of the L2 mutation on the desensitization kinetics. (**D**) Bar graph of the effects of the L2 mutation on the steady state current of the complexes. Filled symbols correspond to the traces shown in (**A**) and (**B**). ***p<0.001, against γ2; ΔΔΔ p<0.001, ΔΔ p<0.1, against γ8. Error bars represent s.e.m. Source data for panels C and D is found in *Table 1–source data 1*. Error bars represent s.e.m.
DOI: https://doi.org/10.7554/eLife.28680.014

Even more striking were results of coexpression of a chimera of γ2 with the GS-linker replacing L2, but harboring the long L1 loop of γ8. This chimera massively slowed entry to desensitization, producing complexes about 10-fold slower than receptors without any TARP ($k_{des}$ = 10 ± 0.5 s$^{-1}$, $n$ = 7; *Figure 3B and C* and *Table 1*), and increased the steady state current during a 500 ms pulse of glutamate (45 ± 3%, $n$ = 7; *Figure 3B and D* and *Table 1*). Making the inverse chimera (L1 from γ2 in γ8, with the GS-linker replacing L2) effectively nullified the modulatory activity of γ8.

The steady-state current was the same magnitude as for receptors that did not have γ8 (5 ± 1%, $n$ = 6; *Figure 3B and D* and *Table 1*), and the rate of entry to desensitization (85 ± 20 s$^{-1}$, $n$ = 6; *Figure 3B and C* and *Table 1*) was closer to that of wild-type GluA2 than for the γ2 L2_GS chimera (see *Table 1*).

Although we performed all measurements at +50 mV, isolating receptors associated to TARPs by selecting for complexes with strong relief of polyamine block, we were concerned that some of the effects that we saw (particularly reduced or absent modulation) could be due to an altered stoichiometry of complexes, perhaps due to poor chimera expression. To assess these possibilities, we measured the G-V relations for all the chimeras and deletion mutants (*Figure 2—figure supplement 1*). Importantly, all mutants gave responses that were strongly reduced in rectification, indicating that complex formation between mutant TARPs and AMPAR subunits was normal. Broadly, each chimera closely followed the polyamine relief induced by the parent TARP, with γ2 chimeras producing populations of receptors that exhibited a greater rectification index than those based on γ8 (*Figure 2—figure supplement 1*). A complication is that modulation of gating and block could depend differently on TARP content within complexes. These results do not address the question of stoichiometry but do strongly suggest that the mutations we made did not change the propensity of γ2 and γ8 to form complexes with AMPAR subunits.

## Superactivation of AMPA-TARP complexes

TARPs induce a subtype-specific superactivation of the GluA2 homomeric receptor. γ8 is a much stronger modifier of this slow gating mode than γ2 (*Kato et al., 2010*; *Carbone and Plested, 2016*). We investigated the role of the extracellular domain in superactivation using the same set of TARP mutants, but using 7 s applications of glutamate to measure the equilibrium level reached following superactivation. Our hypothesis was that the difference in superactivation between γ2 and γ8 would be specified by the sequence element most divergent between these two TARPs, L1.

In the chimeras swapping loop 1 between γ8 and γ2, the results were asymmetric (*Figure 4*). That is, loop 1 from γ8 could transfer the same degree of superactivation to γ2 (L1 γ8 in γ2, 27 ± 6%, $n$ = 10; *Figure 4A and C* and *Table 1*) but the reverse swap could not reduce superactivation to the level of γ2 (L1 γ2 in γ8, 16 ± 1%, $n$ = 16; *Figure 4B and C* and *Table 1*). The reason for this asymmetry became clear when we recorded complexes from which we removed L1 altogether from each TARP. Residual levels of superactivation of 6 ± 2 and 16 ± 3% (for γ2 and γ8, respectively, $n$ = 6; *Table 1*) were still present in the absence of L1. Therefore, although loop 1 can contribute to superactivation, and increase it over baseline levels, it is not the only element of TARPs driving this effect.

Given the residual superactivation that we saw in the absence of loop 1, we reasoned that loop 2 could play a role in receptor superactivation (*Figure 5*). We measured responses to 10 mM glutamate for the L2_GS mutants of γ2 and γ8 and found substantially reduced superactivation (1.3 ± 0.6 and 12 ± 2%, $n$ = 8 and 4, respectively; *Table 1*).

Even more strikingly, the same TARP mutants with loop 1 swapped had a further reduced effect. The loop 1 from γ2 in the L2_GS mutant of γ8 had almost negligible superactivation, reduced by ~15 fold from wild-type γ8, to about 1 ± 0.7% ($n$ = 6; *Figure 5B and C* and *Table 1*). Taking into account the lack of steady-state current, fast desensitization and similar deactivation kinetics to wild-type GluA2 alone that we observed in patches containing complexes of GluA2 with the L1 γ2 in γ8 L2_GS mutant, we classed this chimera as a kinetic null of γ8.

The TARP chimeras that exhibited the least power to slow desensitization kinetics and to stabilize active states were those that replaced charged residues in the L2 segment, and from which we either deleted L1, or included the short loop from γ2. These observations guided our construction of a kinetically-null γ2. We reasoned that a γ2 chimera lacking L1 and with a GS-linker replacing L2 should associate normally with GluA2 but might have no kinetic effect at all on the receptor complexes. Indeed, γ2 ΔL1 L2_GS associated normally into the receptor complex (as assessed by relief of polyamine block, *Figure 6A and B*) but this mutant γ2 was highly deficient in modulating gating of

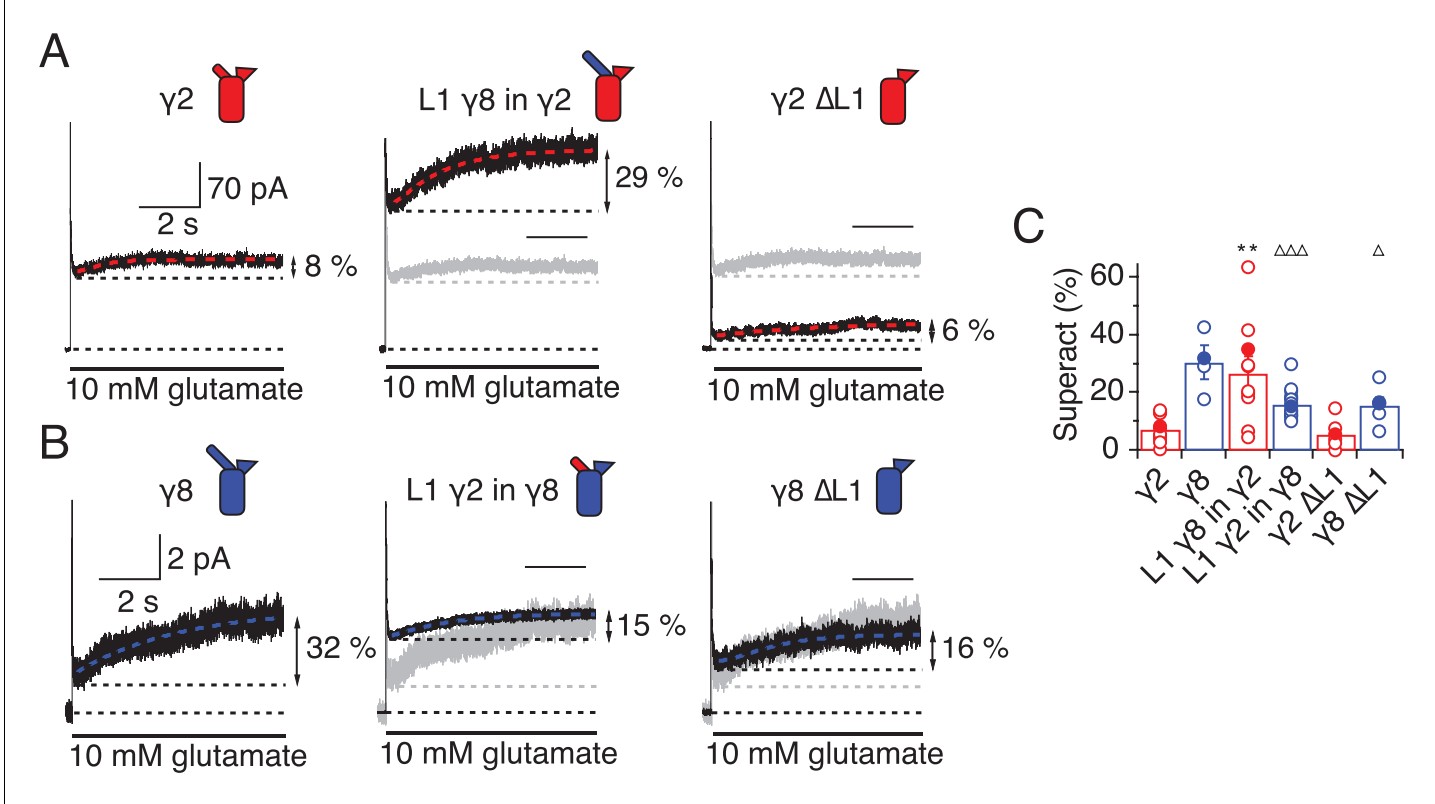

**Figure 4.** L1 modulates the extent of TARP-mediated superactivation. (**A**) Example traces of γ2 wild-type and L1 mutants in response to 7 s application of 10 mM glutamate. During prolonged application of 10 mM Glutamate γ2 induced superactivation of GluA2 receptors, shown as an increase in the steady state current (8% in the example shown, *left panel*). The extent of superactivation was increased by 3-fold when L1 was replaced with that of γ8 (*central panel*). Removing loop1 in γ2 did not affect superactivation much (*right panel*). (**B**) γ8 showed much bigger superactivation than γ2 during long glutamate exposure (*left panel*). Shortening loop 1 by replacing it with that of γ2 or removing it decreased superactivation by 2-fold (*central and right panel*). (**C**) Bar graph summarizing the effects of the loop1 mutations on receptor superactivation. Currents were recorded at +50 mV in the presence of 50 μM spermine in the pipette solution. Filled symbols correspond to the traces shown in (**A**) and (**B**) \*\*p<0.01, against γ2; ΔΔΔ p<0.001, Δ p<0.05, against γ8. Source data for panel C is found in *Table 1–source data 1*. Error bars represent s.e.m.

DOI: https://doi.org/10.7554/eLife.28680.015

GluA2. Superactivation, and the increase in steady state current were absent in these complexes (superactivation = 0%; $I_{ss}$ = 2 ± 1%, $n$ = 4 and 5, respectively; *Figure 6C and D* and *Table 1*). Somewhat surprisingly, the deletion of L1 from γ8 on the L2-GS background retained a larger steady state current than the chimera that included the L1 segment of γ2 ($I_{ss}$ = 5 ± 1% and 10 ± 5%, $n$ = 6 and 5, for L1 γ2 in γ8 L2_GS and γ8 ΔL1 L2_GS, respectively; *Figures 5B and C* and *6E* and F and *Table 1*) and a small superactivation (3 ± 1%, $n$ = 4; *Figure 6A* and *Table 1*).

## L2 controls gating through interaction with linkers proximal to the channel gate

From our models, a range of sites on GluA2 could interact with L1, including the KGK motif in the LBD (*Twomey et al., 2017a*; *Dawe et al., 2016*). Substitutions at L2 of γ2 and γ8 had profound effects on gating of TARP complexes and are well placed to interact with gating machinery (*Figure 1B* and *Figure 1—figure supplement 1B*). Particularly, we expected from our structural models and other available structural data (*Twomey et al., 2016*; *Zhao et al., 2016*) that L2 should interact with the S1-M1 linker and the S2-M4 linker in the AMPA receptor. Previous work has shown the importance of these linkers in glutamate receptor gating (*Balannik et al., 2005*; *Schmid et al., 2007*; *Talukder et al., 2010*). The L2 sequence has an alternating charge motif (see *Figure 1—figure supplement 3*) that is mirrored in two parts of the GluA2 linkers 508QKS510 and 781KEK783.

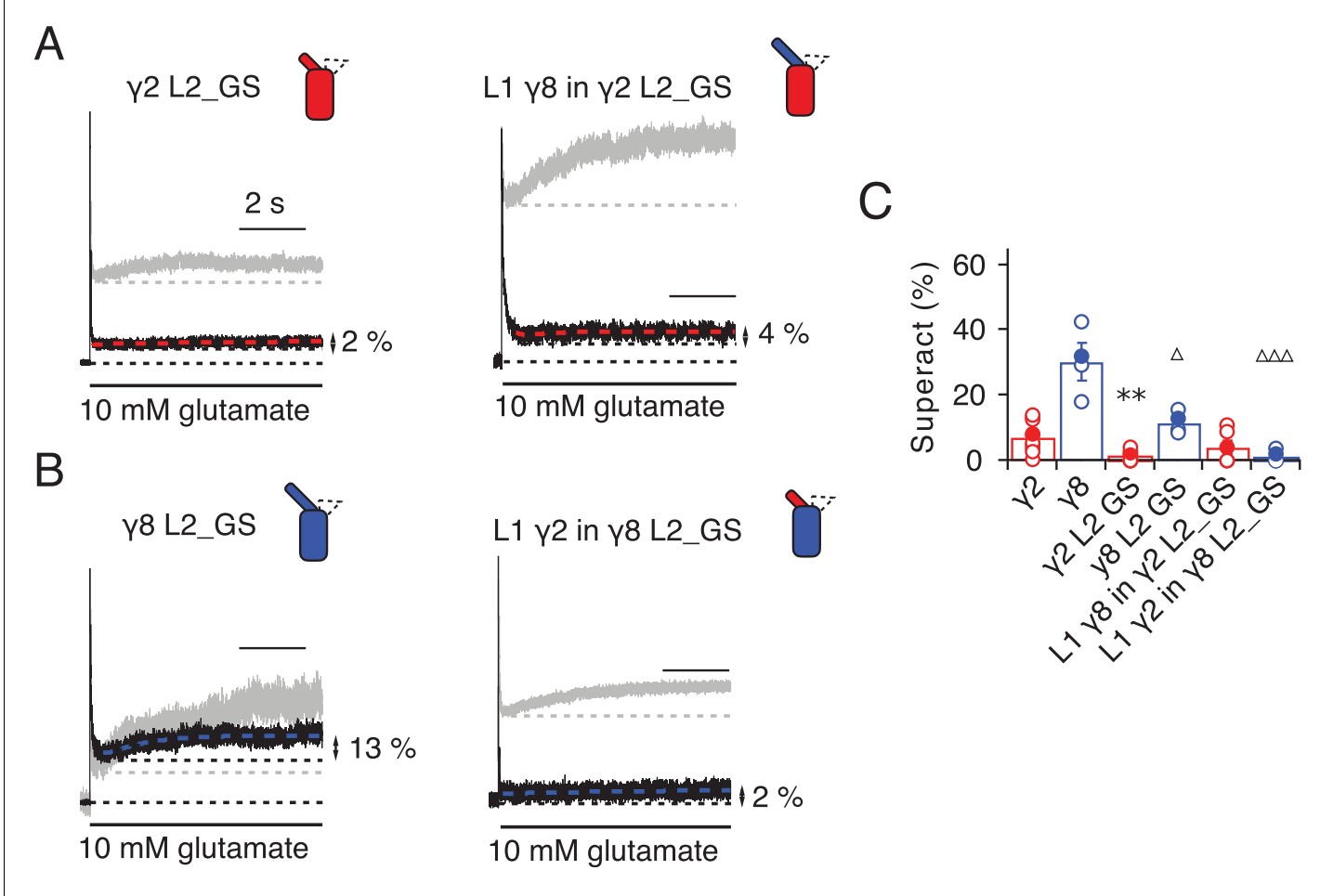

**Figure 5.** Superactivation of γ2 and γ8 L2 mutants. (**A**) Neutralizing L2 from γ2 strongly reduced γ2-mediated superactivation (*left panel*). On this background, L1 from γ8 induced only minimal superactivation (*right panel*). The grey traces represent WT γ2 (*left*) and L1 γ8 in γ2 (*right*). (**B**) Removing L2 in γ8 decreased superactivation 2.5 fold (*left panel*). Introducing L1 from γ2 on this background practically abolished superactivation (*right*). The grey traces represent WT γ8 (*left*) and L1 γ2 in γ8 (*right*). (**C**) Bar graph of the effects of the L2 neutralization and L1 chimeras on superactivation. Filled symbols correspond to the traces shown in (**A**) and (**B**). **p<0.01, against γ2; ΔΔΔ p<0.001, Δ p<0.05, against γ8. Source data for panel C is found in *Table 1–source data 1*. Error bars represent s.e.m.

DOI: https://doi.org/10.7554/eLife.28680.016

These segments are immediately adjacent to the TARP L2 in all four subunits suggesting possible direct electrostatic interactions between opposed charged residues.

Replacement of 508QKS510 (conserved among AMPA receptor subunits) to GAG in the S1-M1 linker (GluA2 508GAG510, *Figure 7A*) produced a GluA2 receptor with normal kinetics and that associated normally with γ2 and γ8 (*Figure 7—figure supplement 1*). Strikingly, in complexes with WT γ2, this mutant phenocopied the neutralizing truncation of L2 in TARPs well (see *Figure 3*), abolishing superactivation and reducing the steady state current (0% and 10 ± 5%, *n* = 3 and 4, for superactivation and $I_{ss}$, respectively; *Figure 7C–E* and *Table 1*). In contrast, a point mutant K509A, also with normal gating (*Figure 7—figure supplement 1*), was more strongly modulated by γ2, providing further indication that a second site was potentially involved (*Figure 7E* and *Table 1*). Our model suggested that the S2-M4 linker of GluA2 was equally well positioned to interact with L2 from γ2. To test the importance of the alternating charges in the S2-M4 linker, we made another triple mutation replacing 781KEK783 (KDK in GluA1, A3, A4) to GSG (GluA2 781GSG783, *Figure 7B*). This mutant again had normal kinetics in the absence of γ2 (*Figure 7—figure supplement 1*), but also exhibited a reduced steady state current and negligible superactivation (10 ± 1% and 2 ± 0.5%, *n* = 9 and 8, for $I_{ss}$ and superactivation respectively; *Figure 7C–E* and *Table 1*). Importantly, the combination of

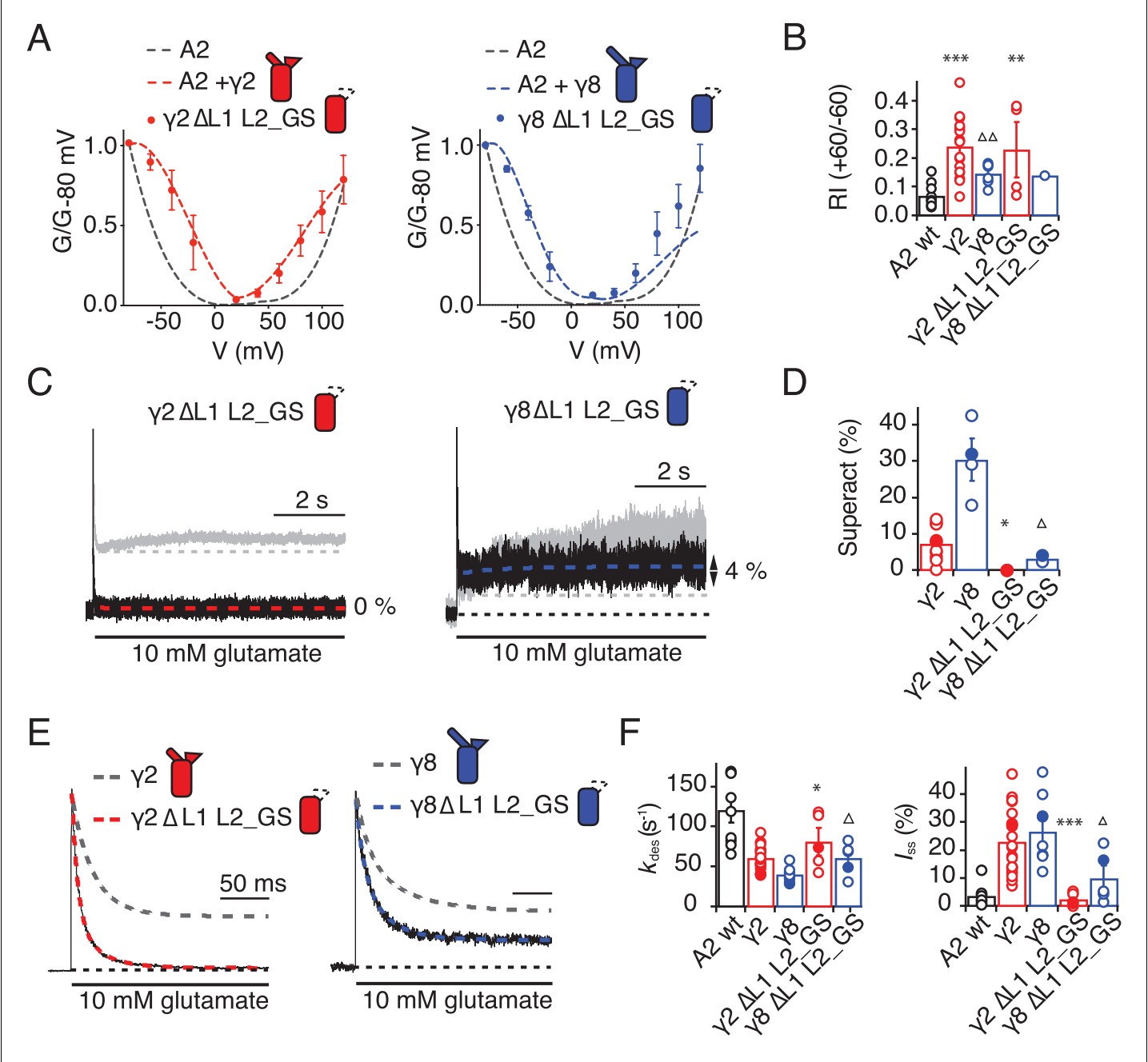

**Figure 6.** Eliminating L1 and L2 removes modulation by γ2. (A) Mutation of both L1 and L2 in γ2 (*left*) and γ8 (*right*) did not change association of TARPs with AMPA receptors, as assessed by the G-V curve. GluA2 WT is shown in grey. (B) Bar graph summarizing the rectification index of the dual loop mutations. (C) Example traces of γ2 ΔL1 L2_GS (*left*) and γ8 ΔL1 L2_GS (*right*) in response to 7 s application of 10 mM glutamate. Corresponding wild-type TARPs are shown as dashed lines. (D) Bar graphs summarizing the effects of the dual loop mutation in γ2 (*red*) and γ8 (*blue*) on superactivation. (E) Representative traces from γ2 ΔL1 L2_GS (*left*) and γ8 ΔL1 L2_GS (*right*) coexpressed with GluA2 in response to a 500 ms pulse of 10 mM Glutamate ($k_{des}$ = 74 and 50 s$^{-1}$ $I_{ss}$ = 1.5% and 16%, respectively). Currents from the parent TARPs are shown in grey. (F) Bar graphs summarizing the effects of the dual loop mutation in γ2 (*red*) and γ8 (*blue*) on desensitization decay and the steady state current. Currents were recorded at +50 mV in the presence of 50 μM spermine in the pipette solution. For panels D, F and G, filled symbols correspond to the traces shown in (C) and (E). ***p<0.001, **p<0.01, *p<0.05, against γ2. Source data for panel B is found in *Figure 6—source data 1*. Source data for panels D, F and G is found in *Table 1–source data 1*. Error bars represent s.e.m.

DOI: https://doi.org/10.7554/eLife.28680.017

The following source data is available for figure 6:

**Source data 1.** Rectification indices for electrophysiological recordings of TARP deletion chimeras.

DOI: https://doi.org/10.7554/eLife.28680.018

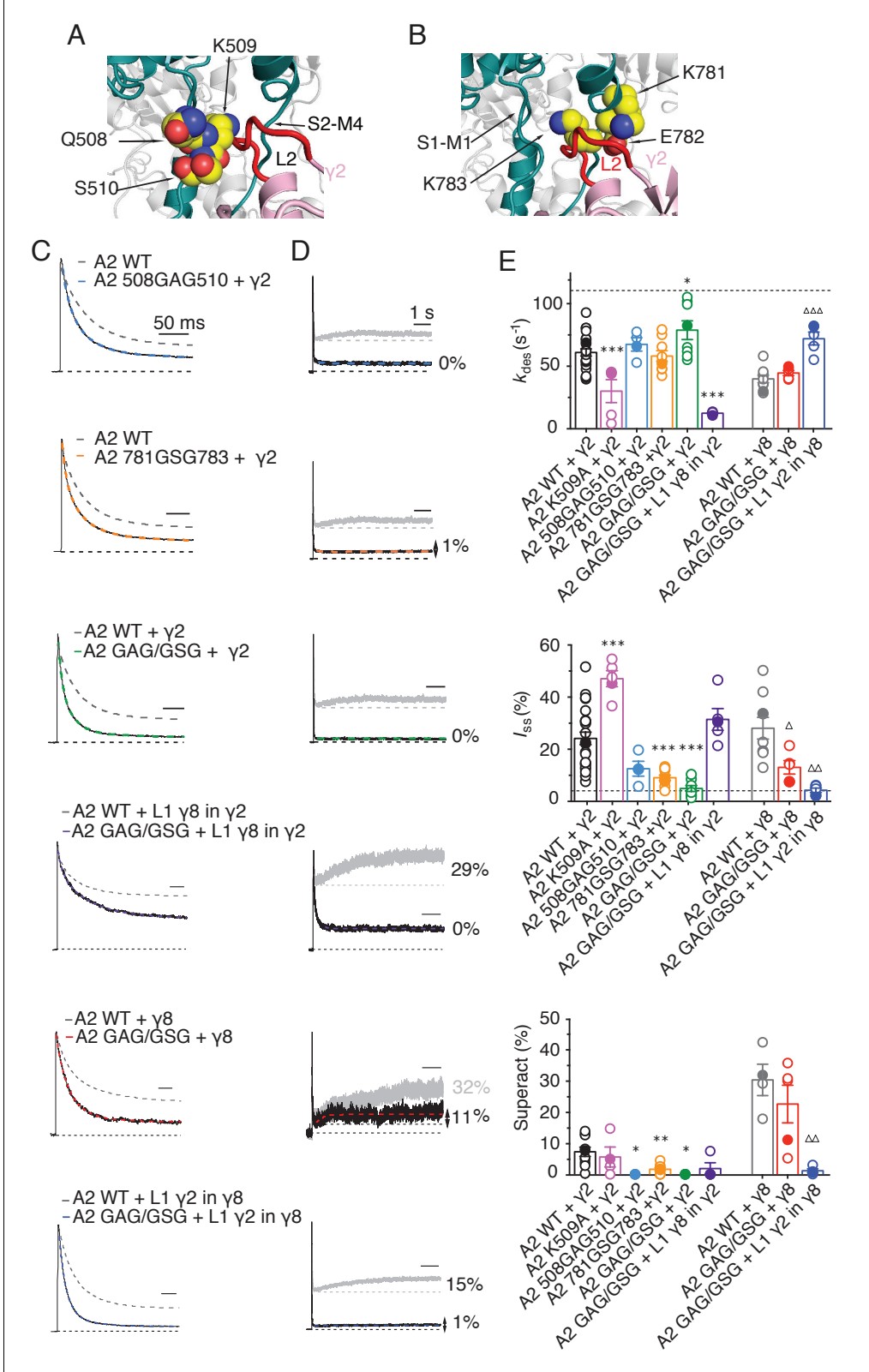

**Figure 7.** The LBD-TMD linkers are the key sites for modulation of AMPA receptor gating by TARPs. (A) Residues in the S1-M1 linker (Gln508, Ser509, and Lys510 represented as yellow atomic spheres) are in close proximity to the L2 of TARPs (L2 of γ2 is shown in red). (B) Residues in the S2-M4 linker (Lys781, Glu782 and Lys783) predicted to interact with L2 are labeled and shown as yellow atomic spheres. (C) Example responses from linker mutants coexpressed with γ2, γ8 and loop 1 chimeras to 500 ms 10 mM Glutamate. (D) Representative responses from linker mutants coexpressed with γ2, γ8

*Figure 7 continued on next page*

eLIFE Research article

Biophysics and Structural Biology | Neuroscience

*Figure 7 continued*

and loop 1 chimeras to a 7 s pulse of 10 mM Glutamate. The extent of superactivation is indicated. (**E**) Bar graphs summarizing the desensitization properties (*top panel*), steady state current (*central*) and superactivation (*bottom*). Colors are as in panel C. Filled symbols correspond to the traces shown in panels C and D; dashed lines represent GluA2 wild-type. ***p<0.001, **p<0.01, *p<0.05, against γ2; ΔΔΔ p<0.001, Δ p<0.05, against γ8. Source data for panel E is found in *Table 1–source data 1*. Error bars represent s.e.m.

DOI: https://doi.org/10.7554/eLife.28680.019

The following source data and figure supplements are available for figure 7:

**Figure supplement 1.** GluA2 linker mutants do not affect receptor kinetics or assembly with TARPs.

DOI: https://doi.org/10.7554/eLife.28680.020

**Figure Supplement 1—source data 1.** Rectification indices for electrophysiological recordings of TARPs with GluA2 mutants.

DOI: https://doi.org/10.7554/eLife.28680.022

**Figure supplement 2.** Thermodynamic coupling analysis for AMPAR linkers and γ2 loop 2.

DOI: https://doi.org/10.7554/eLife.28680.021

these two triple mutants, abolished the entire modulatory effect of γ2 on the AMPA receptor, reducing superactivation and the instantaneous steady-state current to the same level as GluA2 in the absence of TARP (0% and 5 ± 1%, *n* = 4 and 8, for superactivation and $I_{ss}$, respectively; *Figure 7C–E* and *Table 1*). This mutant receptor retained ostensibly normal gating and association to TARPs (*Figure 7—figure supplement 1*), despite the absence of gating modulation.

To test our hypothesis of a direct interaction, we examined receptor responses in the case of both L2 and the receptors linkers being mutated, and constructed thermodynamic cycles to estimate coupling energies for both the steady state current and the superactive current (*Hidalgo and MacKinnon, 1995*). With the caveat that we can at best measure a level of current of about 1% of the peak current, limiting the resolution of the reduction in current upon mutation, we could detect only weak coupling for the steady-state current, but coupling of approximately 4.5 kJ/Mol in the case of superactivation (*Figure 7—figure supplement 2*).

To discern whether the loss of modulation occurred because the linker sites are the primary interaction site, or whether the linkers both interact with TARPs and transmit upstream modulation from sites in the LBD, we assessed modulation by γ8 and related chimeras. The propensity of γ8 to modulate gating of the double linker mutant (GluA2 GAG/GSG) was reduced, but robust superactivation could still be observed (25 ± 5%, *n* = 5, *Figure 7* and *Table 1*). Given this result, which suggested that L1 could still modulate gating of complexes, we hypothesized that the γ2 chimera incorporating the L1 of γ8 should also modulate the double linker mutant. This chimera could not produce superactivating complexes (2 ± 2%, *n* = 4, *Figure 7D and E*, as for the γ2 chimera lacking L2 interactions, L1 γ8 in γ2 L2_GS, *Figure 5A*) but retained the slow desensitizing behavior due to L1 ($k_{des}$ = 12 ± 0.5, *n* = 5, *Figure 7E* and *Table 1*).

In coherence with our previous results, mutation of the GluA2 linkers ablated the effect of the γ8 chimera with L1 from γ2 to modulate the kinetics of complexes, reducing the steady state current and superactivation to the same levels as GluA2 wild-type in the absence of TARP ($I_{ss}$ = 4 ± 1%, superactivation = 1 ± 1%, *n* = 5 and 4, *Figure 7E* and *Table 1*). Therefore, in the absence of the long L1, γ8 fails to modulate GluA2 when the S1-M1 and S2-M4 linker interaction sites are removed (again consistent with its cousin lacking L2 interaction sites, the L1 γ2 in γ8 L2_GS variant; see *Figure 5C*).

Overall, these results indicate that the long loop of γ8 L1 is still able to modulate complexes at extracellular sites with the receptor linker sites disrupted, supporting the idea that the linkers do not function primarily to transduce distant TARP modulation. Rather, the LBD-TMD linkers are the primary modulatory site for both γ8 and γ2. The latter has a short L1 loop, and cannot modulate receptors if the L2 interaction is absent. However, γ8 combines the longer L1 and the L2 site to modulate receptor properties more effectively, in a compound fashion.

## Discussion

The results we present here offer several new insights into TARP function. First of all, extracellular sites account for all the modification of AMPA receptor gating by TARPs. Previous work showed that L1 could transfer aspects of modulation between TARPs, but our experiments indicate that the

second short extracellular segment (L2), which varies strongly in sequence between TARPs, is dominant. Further work will be required to establish the generality of this modulatory mechanism.

Secondly, these same sites do not have any appreciable role in determining assembly of TARP-AMPA receptor complexes. Interactions between transmembrane segments and intracellular regions are responsible for assembly and modulation of polyamine block. We speculate that this division of roles arises because state-dependent gating modification results from transient interactions on a timescale far faster than receptor assembly. Long-lived molecular interactions (~10 s and above) could also underlie modification of gating, but it seems unlikely that fast conformational changes regulate assembly of complexes.

Thirdly, we show that the linkers to the transmembrane domain are key sites for modulation of AMPA receptor gating by auxiliary proteins, and provide insights into the molecular basis of this interaction. The identified sites in the LBD-TMD linkers are highly conserved among all AMPA receptor subunits but not present in the related NMDA and kainate receptors, which are not modulated by TARPs. However, for native receptors, other motifs probably prohibit assembly of TARPs into complexes with these subtypes before the question of modulation is pertinent. Previous work suggested ATD interactions and prominent roles for the LBD in modulation of AMPARs by TARPs, but the interactions we demonstrate here are much more proximal to the channel gate (*Cais et al., 2014*). We could show a very close functional confluence between modifying the receptor itself and modifying each TARP, at an interaction site predicted from structural modeling. The elimination of modulation by nullifying L2 of γ2, or by mutating residues in the LBD-TMD linkers of GluA2, strongly implicates this site as a pivotal interaction underlying modulation. Putative electrostatic interactions posited from structural studies require a large conformational change (between 13 Å and 25 Å depending on the TARP's position in the complex; measured between C-alpha atoms from GluA2 K699 and γ2 D92 in cryo-EM complexes 5kbu and 5kk2, respectively) (*Twomey et al., 2016*; *Zhao et al., 2016*). A key point here is that these interactions are secondary to those involving L2 at the AMPAR linkers. These interactions should occur readily for each auxiliary protein subunit, allowing a maximal 4:4 stoichiometry with minimal conformational change for γ2 (*Figure 8A*) (*Zhao et al., 2016*). For other auxiliary proteins, for example γ8, the stoichiometry of the L2-linker interaction would vary with the number of associated TARPs, but will not be limited by position of the TARP within the complex (*Figure 8B*). Finally, neutralization of the major part of the acidic patch strongly enhanced modulation of gating by γ2, ruling out that negative charges here have a dominant role in modulation.

Fourth, we show that the long extracellular loop 1 of γ8 is a very strong positive modulator of AMPA receptor gating, whose influence is likely held in check by the substoichiometric combination of γ8 with the AMPA receptor (*Hastie et al., 2013*). The subunit γ8 slows receptor desensitization via L1. This loop can produce a profound block of desensitization when transplanted to γ2, and probably interacts state-specifically with the LBD dimer because of its substantial reach (for examples see *Figures 1* and *8*). Previous kinetic measurements suggest that superactivation is adopted by a minor population of receptors in equilibrium with saturating glutamate, speaking in favor of a weak interaction that is boosted by the high effective concentration of L1 close to its site of action in the receptor complex.

Our approach to fit Claudins with modeled loops from TARPs into the best resolution cryoEM reconstruction available (5KK2, [*Zhao et al., 2016*]) has clear implications for modulation. Our model, when compared to the independently derived model of TARP-AMPA modulation (*Twomey et al., 2016*), presents the TARPs oriented at a subtly different angle. Therefore, our model predicted the L2 interaction on the basis of one set of CryoEM data. We could not adequately incorporate the loops and the original structures of the receptor linkers in this model (*Figure 1* and *Figure 1—figure supplement 1*). Whilst this problem could be due to deficits in our model, another explanation is that the linkers (S1-M1 and S2-M4) are disrupted from their basal positions, and that the L2 loop can wedge between them. Upon activation, it is expected that the linkers will move away from the overall pore axis, which could permit further state-dependent interactions (See cartoon in *Figure 8C*).

Future structural studies may permit a more detailed view into the interactions between L2 and the linker domains of AMPAR. Although Claudin structures allowed positioning of auxiliary proteins with high confidence within CryoEM reconstructions, the loops that we have investigated here are not resolved within these structures, possibly because they interact transiently and are otherwise

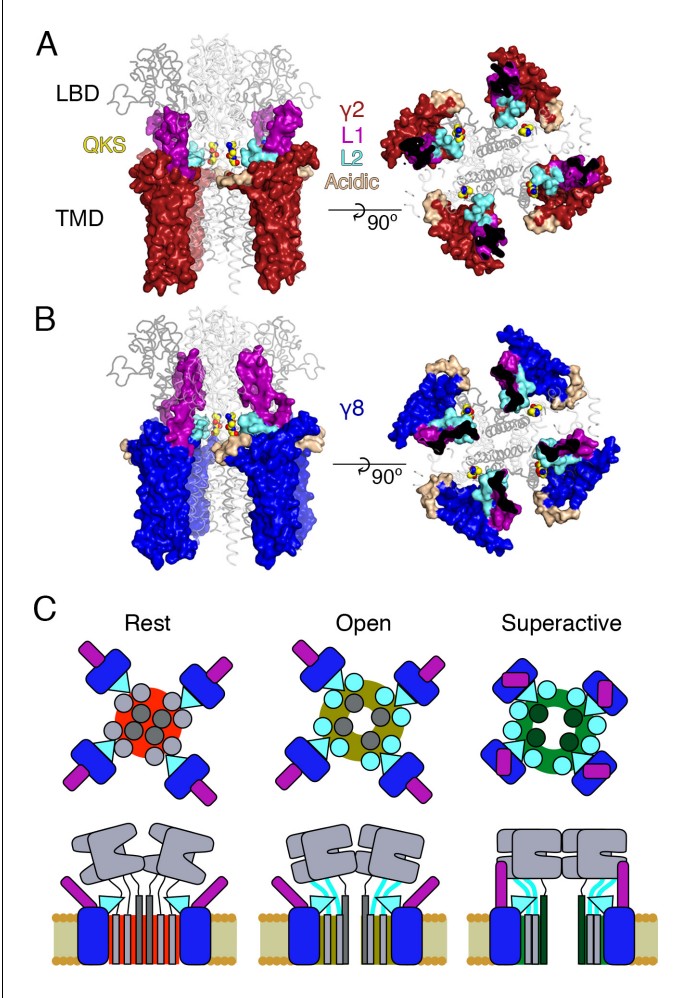

**Figure 8.** Proposed mechanism of AMPA modulation by TARPs. (**A**) Model of a AMPA-γ2 complex in front view (*left*) and top view (*right*). Four molecules of γ2 (*red*) are shown with L1 and L2 colored in magenta and cyan, respectively. L2 is sandwiched between the LBD-TMD connecting linkers of the receptor (grey, amino terminal domains omitted for clarity). The QKS sequence on the S1-M1 linker is shown as yellow atomic spheres. The acidic patch on the β4-TM2 linker is indicated in wheat. (**B**) The model of γ8 (*blue*) shows the similar interactions of L2 (*cyan*). The orientation of the more extensive loop 1 of γ8 is not known, here it is depicted reaching up to the LBD dimer. (**C**) Cartoon model of the proposed AMPA modulation mechanism, taking the example of γ8. The AMPA-TARP complex is shown from top (upper panel) and in side view (lower panel). The receptor is colored in grey (pore forming M3 domain depicted in dark grey). γ8 is colored as in panel B, with the acidic patch omitted. In the resting state (indicated by a red, closed pore) L2 is positioned in close proximity to the LBD-TMD connecting linkers. Once glutamate binds to the LBD, the resulting conformational change is transduced via the LBD-TMD linkers to open the pore (olive green, open state). During this transition L2 could wedge between the S1-M1 and S2-M4 linkers to modulate the receptor gating. The concerted action of L1 and L2 is necessary for superactivation of the receptor (dark green, high open probability state), most likely via L1 to stabilizing the LBDs layer.
DOI: https://doi.org/10.7554/eLife.28680.023

disordered. Although our peptide array suggested that stretches of L1 interact with the LBD, we were not able to obtain co-crystal structures of peptides with monomeric or dimeric forms of the GluA2 LBD. Nonetheless, knowledge of Claudin structures enabled us to make structurally sympathetic substitutions into TARPs for functional experiments that did not disrupt expression or assembly of complexes. These approaches are in contrast with most previous work which simply swapped extracellular portions, including mismatching the folded portions of the TARP extracellular domain. Two observations highlight the importance of sympathetic exchanges. First, some naive deletions would be expected to alter TARP structure. The simple deletion of L2 would severely disrupt the

extracellular domain of γ2 or γ8, because this segment connects structured regions separated by about 10 Å. Second, some deletion chimeras we made retained modulation, with the most striking example being γ8 ΔL1 L2_GS, which retained a substantial steady state current (*Figure 6*). The residual modulation could be related to the presence of a few residues from L1 in the γ8 ΔL1 L2_GS (see *Figure 1—figure supplement 4*). Without maintaining these residues, the chimera did not express. This observation illustrates the sensitivity of domain boundaries in TARPs.

Because our observations suggest that the AMPA receptor linkers are key to TARP modulation, it is likely that chimeric receptors with altered linkers that exhibit constitutive gating are bad reporters of the TARP-GluA modulation, although they clearly delineate assembly motifs (*Ben-Yaacov et al., 2017*). The molecular nature of the interactions we have identified here raise the intriguing possibility that acute disassembly of complexes, rather than modulation, might be the target of recent subtype specific drugs (*Maher et al., 2016*; *Kato et al., 2016*).

Our results allow us to construct a tentative model for the distinct forms of modulation that TARPs produce (*Figure 8C*). The slow increase in glutamate efficacy, which we term superactivation, is specified by the combination of L1 and L2, whereas the basal increase in steady state current arises from L2 alone. We previously modeled the modulatory interaction between TARPs and the AMPA receptor with single conformational change, but did not consider desensitization. The concerted involvement of multiple loops suggests multiple conformational states are required to describe the interaction, most notably in the case of γ8. The greater conformational space that can be explored by loop 1, and its strong connection to superactivation, indicate that these conformational changes could relate to the slow transitions represented in the model of superactivation (*Carbone and Plested, 2016*). In contrast, conformational changes of the linker region of the AMPA receptor upon opening will naturally lead to a state-dependent interaction with L2 of γ2 or γ8, because of the direct proximity. A further level of complexity is that an intact L2 segment is required for the strong superactivation induced by γ8, but is not required at all for slow desensitization behavior that the long L1 loop of γ8 can produce. Because in these experiments, slow desensitization occurs when occupancy of superactive states is low, we can quite reasonably assume that L1 adopts multiple conformations to stabilize separate functional states of the receptor, and that some functional signatures require a concerted action of both loops. Additional stabilization of desensitized states by the variable loop 1 is also likely (*Twomey et al., 2017a*).

This work has produced mutant TARPs and AMPA receptors that both lack modulatory properties, and also those that have greatly enhanced modulation. Both these signatures of activity should be useful tools for investigating TARP action in synapses, including understanding the relative importance of assembly into complexes for anchoring (*Opazo et al., 2010*) as opposed to kinetic modulation, for clarifying the consequences of TARP modulation for short term plasticity (*Devi et al., 2016*), and for better identifying TARPs in ternary complexes with other auxiliary subunits (*Khodosevich et al., 2014*; *Herring et al., 2013*).

## Note

Whilst this manuscript was being revised, an open AMPAR structure in complex with TARP γ2 was released (*Twomey et al., 2017b*). The loops that we analyze here are not resolved, but this structure (like previous closed state structures) places the unstructured L2 in close apposition with the linker regions of the AMPAR, consistent with our idea of state-dependent L2 interactions and modulation.

## Materials and methods

### Molecular biology

We used GluA2 flip receptors, unedited at the pore site (Q-containing) in the pRK vector also expressing eGFP following an internal ribosomal entry site (IRES) sequence. Mouse γ2 was the kind gift of Susumu Tomita and was expressed from an IRES-dsRed construct as previously described (*Carbone and Plested, 2016*). Mouse γ8 (the kind gift of Roger Nicoll) was expressed the same way. Point mutations and chimeras were created by overlap PCR and confirmed by double-stranded sequencing. The construct boundaries of the chimeras used are shown in *Figure 1—figure supplement 4*. Residues in GluA2 were numbered based on the assumption that the signal peptide is 21 residues.

## Patch clamp electrophysiology

Wild type or mutant GluA2 and TARP constructs were co-transfected with PEI in HEK 293 cells (obtained from the Leibniz-Institut DSMZ - Deutsche Sammlung von Mikroorganismen und Zellkulturen GmbH; DSMZ no.: ACC 305). Cell identity was confirmed by immunology and multiplex PCR by DSMZ. 293 cells were tested negative for mycoplasma, both by DSMZ and in-house. The ratios of co-transfection were 1:2 for GluA2-γ2 and 1:5 for GluA2-γ8, up to 2 µg total DNA per 35 mm dish. The same ratios were maintained for all the reciprocal mutants. Cells were supplemented with 40 µM NBQX to reduce TARP-induced cytotoxicity. Recordings were performed 24–48 hr after transfection. The external recording solution contained (in mM): 150 NaCl, 0.1 $MgCl_2$, 0.1 $CaCl_2$ and 5 HEPES, titrated to pH 7.3 with NaOH. The pipette solution contained (in mM): 120 NaCl, 10 NaF, 0.5 $CaCl_2$, 5 $Na_4BAPTA$, 5 HEPES and 0.05 spermine, pH 7.3. 10 mM glutamate was applied to outside-out patches with a piezo-driven fast perfusion system (PI, Germany). In order to isolate currents exclusively mediated by AMPAR-TARP complexes, patches were voltage-clamped at a holding potential of +50 mV. Currents were low-pass filtered at 5 kHz using an Axopatch 200B amplifier (Molecular Devices, U.S.A.) and acquired with Axograph X software (Axograph Scientific, U.S.A.). Typical 10–90% solution exchange times were faster than 300 µs, as measured from junction potentials at the open tip of the patch pipette.

### Data analysis

To measure receptor desensitization we applied 10 mM glutamate for 500 ms. Desensitization rate and steady-state current were then obtained by fitting the traces with a sum of two, and when necessary three, exponentials. Rates constants are expressed as weighted mean of multiple components. Superactivation was measured during a 7 s application of glutamate and was defined as the excess steady-state amplitude following the desensitization trough, normalized to the peak current. The GV relationships were calculated as the ratio between the current amplitude and the voltage step at which it was measured (−80 /+120), normalized against the value obtained at −80 mV. The value at 0 mV was omitted because very close to the reversal potential. The rectification index (RI) was calculated as the ratio between the current amplitude recorded at +60 mV and that recorded at −60 mV. A triple exponential function was used to fit the slowly augmenting current of superactivation measurements. To account for possible variability in the response and expression of the complexes, we tried to record at least 5–6 patches from at least three different transfections for each condition. For experiments with very low success rates (that is, worse than 1 patch in 20 giving an acceptable recording), in the presence of γ8, at least three patches were collected. No data were excluded, except from patches where recordings were unstable, had excessive rundown or solution exchange slower than 0.5 ms as measured after the experiment.

The thermodynamic coupling constant (Ω) was calculated from the ratio of equilibrium constants in the different conditions:

$$\Omega = \frac{E_{wt.wt} \times E_{mut.mut}}{E_{mut.wt} \times E_{wt.mut}}$$

where $E$ is the efficacy of channel opening calculated from the fraction of open receptors ($P_0$):

$$E = \frac{P_0}{1 - P_o}$$

We made the same calculation for the efficacy of receptor superactivation in each condition, using the fraction of superactive receptors in place of $P_0$. The coupling free energy was then calculated as follows:

$$\Delta\Delta G = RT.ln(\Omega)$$

Results are shown as mean ± standard error of the mean (s.e.m.) and statistical significance was assessed with a two-tailed Student's t-test as specified in *Table 1*.

## Protein expression and purification of soluble LBDs

Using the flop isoform of rat GluA2 ligand binding domain (S1S2 fusion) in pET22b vector (kindly provided by E. Gouaux) as a base, we inserted the flip mutations N744T, A745P, N754S, L758V,

and added the C-terminal residues Lys776-Gly779 (GluA2_LBD) and the non-desensitizing mutation L483Y (GluA2_LBD_LY) by overlap mutagenesis. Protein expression and purification was carried out as described previously (*Salazar et al., 2017*). Briefly, monomeric and dimeric (L483Y) LBDs were expressed in *E. coli* Origami B (DE3). Cells were harvested by centrifugation, lysed and subjected to metal affinity chromatography and size exclusion chromatography. Fractions containing the N-terminal His$_8$-tagged protein were pooled and dialyzed against protein buffer (20 mM Tris pH7.4, 150 mM, NaCl, 10 mM glutamate). The purity was determined to >98% by SDS-PAGE analysis.

## Peptide spot array

Peptides covering the extracellular parts of γ2 and γ8 were spotted onto amino modified Whatman cellulose membranes (*Figure 1C* and *Figure 1—figure supplement 1C*) using a fully automatic Spot synthesizer (Intavis, Köln, Germany). The spot array consisted of hexameric overlapping peptides shifted by one residue. Peptide spotted membranes were rinsed with ethanol for 5 min, following three times 10 min washing with TBS and incubation with blocking buffer (Casein Blocking buffer (Sigma B6429), 150 mM Saccharose, in TBS) for 3 hr at RT. The blocking buffer was removed by three wash steps with TBS before the membranes were incubated overnight at 4°C with either 50 μg/ml His-tagged protein (GluA2_LBD or GluA2_LBD_LY) in blocking buffer or blocking buffer only for control (to exclude non-specific binding from the antibodies to the peptides). Membranes were washed three times in TBS and incubated for 1.5 hr at RT with anti-poly_His Antibody (Sigma H1029) diluted 1:6000 in blocking solution followed by three washes (a' 10 min) with TBS. Finally, membranes were incubated for 1.5 hr at RT with HRP-conjugated anti-mouse IgG Antibody (Sigma A5906; 1:1000 dilution in blocking buffer) and washed with TBS (three times a' 10 min). Visualization of protein-binding was carried out using a chemo-luminescence substrate (Pierce ECL, ThermoFisher Scientific) and a Lumi-Imager instrument (Boehringer Mannheim, Germany). Spot-signal intensities were measured in Boehringer Light Units (BLU) and the software GeneSpotter 2.6.0 (MicroDiscovery, Berlin, Germany) was applied for data processing. The peptide spot array is a semiquantitative method to determine binding affinities in the range from nM to higher μM values (the darker the spots the stronger the interaction). As the His-tagged protein is applied in casein containing blocking buffer the detected signals from the His-tagged protein must be higher affine than the unspecific binding from casein to the peptides. Hits from peptides located within β-sheets were taken to be false positives, because when isolated these peptides likely form unphysiological β-sheets in a non-specific manner with existing structures in the GluA2 LBD. To have an idea about reproducibility of this assay, we performed it twice with comparable results (source data is provided). The negative control (incubation of the membrane in blocking buffer only) showed no signal, indicating no unspecific binding of the anti-poly His antibody to the peptides.

## Structural modeling

Initial γ2 and γ8 models were generated based on the crystal structure of claudin15 (PDB code: 4P79) using the SWISS-MODEL (*Arnold et al., 2006*) and ProtMod server (part of the FFAS server, [*Jaroszewski et al., 2011*]). Both models were incomplete (either lacking linker structures or failing to correctly trace transmembrane helix 3, TM3). Thus, we used COOT (version 0.8.7) to superpose the two generated models and to build the final model with an intact helix 3 and plausible extracellular loops 1 and 2. Superposing our final TARP models onto the γ2 molecules present in the AMPA-TARP cryo-EM structure (PDB code: 5KK2) in PyMOL (v1.6) yielded in the AMPA-TARP complexes shown in our Figures. The two extreme different possible orientations of Loop 1 were modeled using COOT. The TARP models were validated using MolProbity (*Chen et al., 2010*). Unfortunately the LBD to TMD connecting linkers (S1-TM1 and S2-TM4) are not resolved in the AMPA-TARP cryo-EM structure. To better understand the Loop 2 participation in AMPA receptor regulation we used the crystal structure of GluA2 (PDB code: 3KG2) with resolved linkers and superposed it onto the receptor of our AMPA-TARP complex model (*Figure 1—figure supplement 2*). As the side chains of the possible interacting residues (507-QKS-510, 781KSK-783) located in the LBD-TMD linkers were not resolved in 3KG2 we modeled the most likely side chain conformations of these residues (*Figure 7A and B*). All figures were prepared with PyMOL or IGOR Pro. Sequence alignment was done with MultAlin (*Corpet, 1988*).

## Acknowledgements

This work was funded by the Deutsche Forschungsgemeinschaft (DFG) – FOR 2518 ('DynIon', to AJRP), the DFG Cluster of Excellence 'NeuroCure' (DFG EXC-257, to ALC), and an Erwin-Schrödinger Postdoctoral Fellowship (J3682-B21) of the Austrian Science fund (FWF, to CE). AJRP is a Heisenberg Professor of the DFG. We thank Marcus Wietstruk and Ronny Schäfer for technical assistance.

## Additional information

### Funding

| Funder | Grant reference number | Author |
|---|---|---|
| Deutsche Forschungsgemeinschaft | Fellowship and studentship Cluster of Excellence NeuroCure EXC-257 | Anna L Carbone |
| Austrian Science Fund | Erwin-Schrödinger Postdoctoral Fellowship J3682-B21 | Clarissa Eibl |
| Deutsche Forschungsgemeinschaft | Research Group Dynion FOR 2518 | Andrew JR Plested |
| Deutsche Forschungsgemeinschaft | Heisenberg Professor | Andrew JR Plested |

The funders had no role in study design, data collection and interpretation, or the decision to submit the work for publication.

### Author contributions

Irene Riva, Formal analysis, Investigation, Visualization, Writing—review and editing; Clarissa Eibl, Conceptualization, Formal analysis, Funding acquisition, Investigation, Visualization, Methodology, Project administration, Writing—review and editing; Rudolf Volkmer, Conceptualization, Formal analysis, Methodology; Anna L Carbone, Conceptualization, Formal analysis, Supervision, Funding acquisition, Investigation, Methodology, Project administration, Writing—review and editing; Andrew JR Plested, Conceptualization, Formal analysis, Supervision, Funding acquisition, Visualization, Methodology, Writing—original draft, Project administration, Writing—review and editing

### Author ORCIDs

Anna L Carbone [ORCID] http://orcid.org/0000-0001-9746-2778
Andrew JR Plested [ORCID] http://orcid.org/0000-0001-6062-0832

### Decision letter and Author response

Decision letter https://doi.org/10.7554/eLife.28680.025
Author response https://doi.org/10.7554/eLife.28680.026

## Additional files

### Supplementary files

• Transparent reporting form
DOI: https://doi.org/10.7554/eLife.28680.024

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
