## [Decision Letter]

Thank you for submitting your article "Control of AMPA receptor activity by the extracellular loops of auxiliary proteins" for consideration by *eLife*. Your article has been reviewed by three peer reviewers, and the evaluation has been overseen by Kenton Swartz as the Reviewing Editor and Gary Westbrook as the Senior Editor. The following individuals involved in review of your submission have agreed to reveal their identity: Lonnie Wollmuth (Reviewer #1); Hiro Furukawa (Reviewer #2).

The reviewers have discussed the reviews with one another and the Reviewing Editor has drafted this decision to help you prepare a revised submission.

Summary:

A series of recent and exciting publications have explored structural and mechanistic interactions between AMPA receptors and TARPs, a family of accessory subunits that modulate gating of the receptor. In the present manuscript, the authors use chimeras as well as site specific mutations to identify interactions of the extracellular regions of gamma2 and gamma8, L1 loop and L2 loop, with AMPA receptor subunits. They find that L1 loop, which interacts predominantly with LBD, modulates desensitization properties. On the other hand, L2 loop, which presumably interacts with the LBD-TMD linkers, has much more notable effects on desensitization and superactivation. The authors are also able to generate what they call kinetic null constructs – constructs that show no effect on gating – for either the extracellular loops of gamma2 or gamma8 as well as for LBD-TMD linkers. This is a particularly interesting discovery. Overall, the manuscript is well-written and for the most part data are presented clearly. The results and conclusions are extremely interesting. There was concern with the model because it wasn't clear how it was generated and it has not been validated as far as we could see. The following are essential revisions that the authors should address.

Essential revisions:

1) The value and validity of the model presented in Figure 1 is unclear. The authors need to expand on the description of how the model was generated and describe what kind of constraints were added to the loop region. The fact that there is no density in cryo-EM may indicate that the loop is disordered or has too many conformations to be tracked. We think that the quality of the manuscript, which is otherwise excellent, is diminished by this casual web-based model and the extent to which the authors build hypotheses around it. The authors should also try to validate the model, perhaps by calculating rmsd for bond length and bond angle using programs such as Procheck and Molprobity, keeping in mind that this analysis only tests the geometrical parameters of created models and cannot validate the existence of the loop conformations that authors show in their manuscript. We also request that the authors revise the manuscript to diminish the extent to which the model is used to frame hypotheses and perhaps simply state that the "loop is long enough to perhaps transiently interact with AMPAR LBD."

2) The experiments shown in Figure 1 need to be better introduced and documented by the authors. It is very difficult to know what the authors did here experimentally. Did the authors detect bound LBD tagged with poly histidine tag? It needs some explanations in the figure legend or in the Materials and methods section. The authors also need to justify what they think this assay can tell us. How strong or specific must the interactions be to be detected here? There are multiple controls in these experiments, and the authors should walk the reader through them and explain how they constrain their results.

---

## [Author Response]

Essential revisions:1) The value and validity of the model presented in Figure 1 is unclear. The authors need to expand on the description of how the model was generated and describe what kind of constraints were added to the loop region. The fact that there is no density in cryo-EM may indicate that the loop is disordered or has too many conformations to be tracked. We think that the quality of the manuscript, which is otherwise excellent, is diminished by this casual web-based model and the extent to which the authors build hypotheses around it. The authors should also try to validate the model, perhaps by calculating rmsd for bond length and bond angle using programs such as Procheck and Molprobity, keeping in mind that this analysis only tests the geometrical parameters of created models and cannot validate the existence of the loop conformations that authors show in their manuscript. We also request that the authors revise the manuscript to diminish the extent to which the model is used to frame hypotheses and perhaps simply state that the "loop is long enough to perhaps transiently interact with AMPAR LBD."

The reviewers are right, we should have explained better how this model was validated and checked against existing data. We extended the description on how this model was generated, and included values from the model validation with MolProbity in the second paragraph of the Results subsection “A model of auxiliary protein interactions”. We did not add additional constraints to the loop regions other than those that are used by the model-building servers. It is true that we do not have good information about where loop 1 is. Even in the context of the latest structural data, we have very little information about potential interactions (and nothing for gamma-8). The major aim of the model was to ensure that we know where loop 1 is not. Principally, with this simple model we could limit the physically plausible interacting sites to the LBD, therefore legitimising our peptide screen. Apart from this, we simply took two extreme positions that were possible for the loop 1, taking into account data from Bowie’s lab that the “KGK motif” could be involved somehow. We try to make this rationale clearer in the text now.

2) The experiments shown in Figure 1 need to be better introduced and documented by the authors. It is very difficult to know what the authors did here experimentally. Did the authors detect bound LBD tagged with poly histidine tag? It needs some explanations in the figure legend or in the Materials and methods section. The authors also need to justify what they think this assay can tell us. How strong or specific must the interactions be to be detected here? There are multiple controls in these experiments, and the authors should walk the reader through them and explain how they constrain their results.

We apologise for the lack of clarity in the description here. We added more information in the Materials and methods on the specificity and binding strength that is expected in the case of a positive result in this assay (subsection “Structural modeling”). The negative control is also explained. The results were rewritten to better explain that we were probing direct interactions between the receptor LBD and the extra-cellular segment of TARPs. Further changes were made to the figure legend to guide the reader through the experiment. We hope that the text is now clearer, and appropriately describes what can be learned from this experiment.